# Long-Chain Polyisoprenoids Are Synthesized by AtCPT1 in *Arabidopsis thaliana*

**DOI:** 10.3390/molecules24152789

**Published:** 2019-07-31

**Authors:** Przemyslaw Surowiecki, Agnieszka Onysk, Katarzyna Manko, Ewa Swiezewska, Liliana Surmacz

**Affiliations:** Institute of Biochemistry and Biophysics, Polish Academy of Sciences, Pawinskiego 5a, 02-106 Warsaw, Poland

**Keywords:** polyisoprenoids, polyprenols, dolichols, *cis*-prenyltransferase, protein *N*-glycosylation, *Arabidopsis thaliana*, T-DNA insertion mutant, *rer2*Δ and *srt1*Δ *Saccharomyces cerevisiae* mutant

## Abstract

Arabidopsis roots accumulate a complex mixture of dolichols composed of three families, (i.e., short-, medium- and long-chain dolichols), but until now none of the *cis*-prenyltransferases (CPTs) predicted in the Arabidopsis genome has been considered responsible for their synthesis. In this report, using homo- and heterologous (yeast and tobacco) models, we have characterized the *AtCPT1* gene (At2g23410) which encodes a CPT responsible for the formation of long-chain dolichols, Dol-18 to -23, with Dol-21 dominating, in Arabidopsis. The content of these dolichols was significantly reduced in *AtCPT1* T-DNA insertion mutant lines and highly increased in *AtCPT1*-overexpressing plants. Similar to the majority of eukaryotic CPTs, AtCPT1 is localized to the endoplasmic reticulum (ER). Functional complementation tests using yeast *rer2*Δ or *srt1*Δ mutants devoid of medium- or long-chain dolichols, respectively, confirmed that this enzyme synthesizes long-chain dolichols, although the dolichol chains thus formed are somewhat shorter than those synthesized in planta. Moreover, AtCPT1 acts as a homomeric CPT and does not need LEW1 for its activity. AtCPT1 is the first plant CPT producing long-chain polyisoprenoids that does not form a complex with the NgBR/NUS1 homologue.

## 1. Introduction

Lipids, beside nucleic acids and proteins are vital constituents of all living cells. They have many biological functions in cells as integral components of biological membranes, regulators of metabolic processes or signaling molecules. Many of them are essential for plant survival. Hydrophilic isoprenoids and their derivatives play important role in photosynthesis (plastoquinone, chlorophylls), respiration (ubiquinone), hormonal regulation of metabolism and membrane stabilization (sterols), regulation of growth and development (giberellic acid, abscisic acid, brassinosteroids) and defense against oxidative stress (tocopherol, carotenoids, ubiquinol, plastoquinol). Among these are polyisoprenoids playing a crucial role in posttranslational protein modifications as indispensable saccharide carriers in the biosynthesis of glycosyl-phosphoinositol (GPI) anchor, and protein *N*-, *O*- and *C*-glycosylation [1,2,3] as well as donors of isoprenoid groups in protein prenylation [4]. Moreover, they are involved in cell adaptation to adverse environmental conditions [5], possibly via modulation of the properties of biological membranes—by increasing their permeability and fluidity [6]. Plant polyprenols accumulated in chloroplasts determine the fluidity of thylakoid membranes that affect photosynthetic efficiency [7]. Despite the fact that they were identified already in the 1960s, and regardless of extensive studies, relatively little is still known about the regulation of polyisoprenoid synthesis. 

Polyisoprenoid alcohols have been found in the cells of all living organisms from bacteria to human. These linear polymers, built from five to more than 100 isoprene units, are divided into two subgroups of compounds with either a hydrogenated (dolichols) or an unsaturated (polyprenols) double bond in the α-unit (Figure 1). Dolichols are mainly present in animal cells, yeasts and plant roots whereas polyprenols in bacteria and plant photosynthetic tissues. In eukaryotic cells polyisoprenoids frequently occur as a mixture of homologues named ‘family’ with one dominant component. Several plant species accumulate more complex mixtures containing two or three ‘families’. 

The biosynthesis of the polyisoprenoid skeleton in plants is a complex process and occurs in three steps. At first, the five carbon molecules of IPP (isopentenyl diphosphate) and DMAPP (its allylic isomer dimethylallyl diphosphate) are synthesized via the cytoplasmic mevalonic acid (MVA) and the plastidial 2-methyl-d-erythritol-4-phosphate (MEP) pathways. Next, *cis*-prenyltransferases (CPTs) catalyze the formation of a linear polymer, usually 70–130 carbons in length, by sequential condensations of IPP to short-chain oligoprenyl precursors, most often farnesyl diphosphate (FPP) or geranylgeranyl diphosphate (GGPP) [6]. Eukaryotic CPTs form a broad range of products of variable chain-length, both short and long-chain [8]. In all cells CPTs produce prenyl diphosphate that in some tissues finally is converted to dolichol due to the activity of polyprenol reductase [9,10]. 

The prenyl diphosphate biosynthetic pathway in eukaryotic cells requires the interaction of CPT with an accessory CPT-binding protein for determination of polyprenol chain length and examples of such complexes are human (hCIT/NgBR) [11], yeast (Rer2/Srt1/NUS1) [12] and some plants, including tomato (SlCPT3/SlCPTBP) [13], lettuce (CPT3/CPTL2) [14], Arabidopsis (CPT3/LEW1) [15], ginseng (CPT1/PgCPLT2) [16] and guayule (PaCPT1-3/PaCBP) [17]. 

Plant cells usually express a family of genes encoding potential CPTs while the capacity to synthesize polyisoprenoid chains is confirmed for only some of them. Thus seven genes encoding CPT are identified in tomato (*SlCPT1*–*SlCPT7*) [8], two in *Hevea brasiliensis* (HRT1 and HRT2) [18], three in *Taraxacum koksaghyz* (TkCPT1–TkCPT3) [19] and one (LAA66) in *Lilium longiflorum* [20]. The Arabidopsis genome contains nine putative *cis*-prenyltransferase encoding genes (AtCPT1 to AtCPT9) that are expressed in tissue-specific manner. Three of them, AtCPT1, -6 and -9 are exclusively expressed in roots [21]. Simultaneously several dolichols are identified in this tissue (hairy roots): a single dolichol Dol-7 and three families of dolichols, with Dol-13, Dol-16 and Dol-21/23 dominating [22]. So far, only one root CPT, AtCPT6 involved in Dol-7 biosynthesis has been characterized in planta [23,24]. The enzymes responsible for the biosynthesis of the remaining dolichols in *Arabidopsis thaliana* roots have not yet been described. AtCPT1 (At2g23410) has been identified [25,26] but its location and function in the plant cell remains unknown. In contrast to roots the profile of polyisoprenoids is much simpler in the leaves; only one family of polyprenols (Pren-10 dominating) together with the expression of AtCPT7 is detected [7].

In the present study, an activity of AtCPT1 (At2g23410) was analyzed in hetero- and homologous systems. Both, transient expression in *Nicotiana benthamiana* leaves and stable overexpression in *Arabidopsis thaliana* revealed that AtCPT1 is localized to the endoplasmic reticulum (ER) and catalyzes the synthesis of a family of long-chain dolichols ranging from 19 to 24 i.u., with Dol-21/23 dominating. Moreover, comparing to wild type plants the content of these dolichols was highly increased in the roots upon *AtCPT1* overexpression and significantly lowered for *AtCPT1* T-DNA insertion mutants. Furthermore, upon expression in yeast cells, this plant CPT was involved in the synthesis of a family of polyisoprenoids with chains somewhat shorter than those synthesized in planta (i.e., a mixture of dolichols composed of 16 to 20 i.u.), with Dol-17/18 dominating. Taken together, data presented here document that AtCPT1 catalyzes the formation of long-chain polyisoprenoid products in plant roots. 

## 2. Results and Discussion

### 2.1. AtCPT1 Synthesizes the Family of Long-Chain Dolichols in Planta

To characterize the role of AtCPT1 in dolichol biosynthesis in Arabidopsis roots, we obtained three transferred DNA (T-DNA) inserted *AtCPT1* mutants: *cpt1-1* (SALK_038151+/−), *cpt1-2* (SALK_032276+/−, previously described as a T-DNA insertion mutant of *At1G11755* (LEW1) gene [27]) and *cpt1-3* (SALK_100795−/−) and AtCPT1 overexpressing line *CPT1-OE* in the Columbia (Col-0) background. We confirmed that all of the above mutant lines contained the T-DNA insertion sites in *At2G23410* gene by PCR with T-DNA-specific and flanking primers and by sequencing PCR products. *cpt1-1* line carried a T-DNA insertion in the promoter, *cpt1-2*—in the second exon and *cpt1-*3—in the 3′UTR region (Figure 2A). It appeared impossible to obtain homozygous T-DNA insertion plants of *cpt1-1* and *cpt1-2* mutant lines which might suggest that complete disruption of *AtCPT1* is lethal. 

Quantitative Real Time PCR analysis of AtCPT1 expression in the roots of mutant lines indicated that *AtCPT1* transcripts were significantly reduced, by approximately 80–95%, in all insertion lines and significantly (100-fold) elevated in the *CPT1-OE* line relative to wild type levels (Figure 2B). *AtCPT1* deficiency or overexpression did not affect the expression of the remaining *AtCPT* genes expressed in roots (Appendix A). Residual *AtCPT1* expression was also noted in the leaves of the wild type (WT) plants while for the leaves of the transgenic *CPT1-OE* plants it was increased 800-fold relative to the WT control (Appendix A). 

Analysis of the content of total polyisoprenoids confirmed that roots of WT plants accumulated a three family mixture of dolichols: short-chain dolichols (Dol-13 dominant), medium-chain dolichols (Dol-16) and long-chain ones (Dol-21) accompanied by respective polyprenols (Figure 2C). The content of long-chain polyisoprenoid family with Pren/Dol-21 dominating was significantly decreased in all of the T-DNA mutant lines (*cpt1-1*, *cpt1-2* and *cpt1-3*) to 30%, 42 and 52% of the WT level, respectively, while it was significantly increased (8-fold) in *CPT1-OE* plants (Figure 2C,D). *AtCPT1* deficiency or overexpression did not affect the content of the remaining dolichols in Arabidopsis roots (Appendix A). In line with an increased transcript level, a large amount of long-chain dolichols was also detected in leaves of *AtCPT1-OE* plants (Appendix A) while those of wild type plants did not accumulate products of such length (Appendix A). It is worth noting that the dominating homologue in this case was Dol-23.

Likewise, a qualitative analysis of polyisoprenoids extracted from *Nicotiana benthamiana* leaves transiently expressing *AtCPT1* revealed the presence of an additional dolichol family ranging from 19 to 24 i.u., with Dol-23 dominating, in comparison to mock leaves (Appendix A).

This discrepancy in dominating homologue between long-chain dolichols accumulated in roots and leaves upon *AtCPT1* expression might result from divergent mechanisms responsible for regulation of AtCPT1 activity/specificity on these tissues, moreover it might be due to differences in the availability of a precursor. In line with this, we previously observed that in Arabidopsis hairy root cultures obtained from callus the profile of long-chain polyprenols and dolichols was variable and dependent on the sugar type and its concentration in the growth medium [22]. Also in *S. cerevisiae* cells upon starvation (glucose vs. ethanol) the shift of dominating long-chain dolichol from Dol-21 to Dol-23 was observed [28]. 

Taken together, these results clearly indicate that AtCPT1 is responsible for the synthesis of long-chain polyisoprenoids in plants similarly to Srt1 in yeast [29,30]. Alike Srt1 [28] polyprenol products of AtCPT1 are not efficiently reduced to dolichols.

### 2.2. Shortage of Long-Chain Dolichols Leads to the Growth Defects

The role of long-chain dolichols in plants is still unknown. In Arabidopsis hairy roots their profile is modulated by the sugar type and availability [22]. In *Saccharomyces cerevisiae* long-chain dolichols synthesized by Srt1 are involved in spore wall formation [31].

All three *cpt1* mutants analyzed by us exhibited growth defects. Comparing to the wild type and *CPT1-OE*, the *cpt1* mutants seedlings showed extremely stunted growth and shorter root length (Figure 3). In tomato the reduced expression of SlCPT3, responsible for medium-chain dolichol biosynthesis, led to a pleiotropic phenotype, which included mottled, wilted leaves and stunted growth [13]. In contrast, none of the so far characterized Arabidopsis CPT loss-of-function mutants, AtCPT6 [24] or AtCPT7 [7] (synthesizing Dol-7 or Pren-9 to -11, respectively) showed any phenotypic aberrations. Similarly, the yeast *srt1*∆ mutant has no clear morphological phenotype, although it lacks the outer spore wall layer due to a defect in chitin synthesis [31]. Observed disturbances in the growth of *cpt1* mutants require explanation and will be addressed in future work. 

### 2.3. AtCPT1 Is Localized to the ER

The eukaryotic *cis*-prenyltransferases implicated in dolichol biosynthesis are mainly located in the endoplasmic reticulum (ER) with the exception of yeast Srt1 producing long-chain dolichols that is localized to lipid bodies [30,31]. Membrane topology analysis (TMHMM server) predicted one transmembrane domain in the AtCPT1 amino acid sequence suggesting that this CPT can be an integral organellar membrane protein. Moreover, computational tools to predict subcellular location of proteins (e.g., BaCelLo, TargetP) indicated AtCPT1 localization in the ER/Golgi network. To determine the subcellular localization of AtCPT1, AtCPT1:G3GFP construct, encoding a C-terminal fusion with fluorescent protein was transiently co-expressed with plant organelle markers (mCherry or CFP fusions) in *Nicotiana benthamiana* leaves. Confocal laser microscopy clearly showed that GFP signal fully overlapped with that of CFP fluorescence of the ER marker (Figure 4A). The same fluorescence pattern was observed in the parallel experiment when the ER-mCherry marker was used (Figure 4B). On the contrary, no overlap of AtCPT1-GFP and Golgi-mCherry marker was noted (Figure 4C). 

Despite the fact that AtCPT1 synthesizes dolichols of similar length to yeast Srt1, its subcellular location is different from the yeast CPT.

### 2.4. AtCPT1 Complements the Function of Yeast Rer2 cis-Prenyltransferase

To confirm whether AtCPT1 can fulfill its function in yeast, the full length *AtCPT1* cDNA was introduced into the thermosensitive *rer2*Δ yeast strain devoid of one of the two yeast CPTs, Rer2 and its ability to suppress the growth and protein *N*-glycosylation defects was studied. Transformation with *AtCPT1* suppressed the growth defects of the *rer2*Δ mutant at 37 °C while the empty vector did not restore growth at the non-permissive temperature. Wild type yeast exhibited normal growth in these unfavorable conditions (Figure 5A). This result is in agreement with observations described previously [25].

To verify that the rescue of *rer2*Δ growth by AtCPT1 was due to the complementation of the defect in CPT activity, the effect of *AtCPT1* expression on the glycosylation status of lysosomal protein carboxypeptidase Y (CPY) was examined. WT yeast cells contain only a mature fully glycosylated form of CPY with four *N*-glycan chains in contrast to *rer2*Δ mutant where both the mature and hypoglycosylated CPY variants are present (Figure 5B). Expression of *AtCPT1* fully restored normal *N*-glycosylation of CPY. In conclusion, experiments performed using *rer2*Δ mutant clearly show that AtCPT1 can functionally replace the yeast CPT Rer2. 

Finally, to confirm the role of AtCPT1 in long-chain dolichol biosynthesis we analyzed the polyisoprenoid profile of yeast strains, *rer2*Δ and *srt1*Δ devoid of one of the two yeast CPTs, Rer2 or Srt1, respectively expressing *AtCPT1.* HPLC/UV analysis revealed that both *rer2*Δ/*AtCPT1* and *srt1*Δ/*AtCPT1* yeast transformants accumulated an additional family of dolichols composed of 15 up to 20 i.u., with Dol-17/18 dominating (Figure 5C). This result shows that upon expression in yeast cells, AtCPT1 is able to synthesize a family of polyisoprenoids although their chains are shorter than those synthesized in planta. This observation is consistent with the result observed previously upon in vitro assay where a membrane fraction prepared from yeast transformed with *AtCPT1* (called *DPS*) was used as a source of enzyme. In these experimental conditions, a mixture of dolichols ranging in chain length from C75 to C95 was formed [25]. Moreover, similar lipid products were obtained in vitro when recombinant AtCPT1 (ACPT) protein purified from *E. coli* was used [26]. This interesting discrepancy requires explanation and suggests that long-chain dolichol biosynthetic machinery in Arabidopsis is complex and possibly other plant regulatory protein(s) is necessary to modulate elongation of polyisoprenoid chains. One such candidate is Arabidopsis LEW1, a homolog of human NgBR and yeast Nus1. Nus1 is compatible with heterologous CPT to form an enzymatically active complex (e.g., Rer2 of *S. pombe* and human CIT), but the dominant polyisoprenoid products synthesized by the hybrid enzymes are different than those produced by the wild type enzyme complex [12].

To explain whether AtCPT1 requires activity of CPT-binding protein for formation of long-chain dolichols we co-expressed the plant NgBR/NUS1 homologue: *LEW1* and *AtCPT1* in *rer2*Δ mutant cells. HPLC/UV analysis revealed that both yeast transformants expressing either *LEW1/AtCPT1* or an individual *AtCPT1* accumulated the same family of dolichols with dominating Dol-17/18 while no such products were observed upon expression of *LEW1* or transformation with an empty vector (Figure 5C). This might suggest that LEW1 does not serve as a regulatory protein for AtCPT1 and is consistent with the concept that CPTs with an RXG motif do not need a NgBR/NUS1 homologue protein for their activity [32]. Experimental data and in silico analysis (Figure 6) revealed that AtCPT1, similarly to AtCPT6 and -7 [7,24], was classified into a group of homomeric CPT in contrast to heteromeric CPTs (e.g., AtCPT3) which require the interaction with LEW1 for polyisoprenoid chain biosynthesis [15]. Bacteria possess homomeric while animals and yeast possess heteromeric CPTs; interestingly both types of CPTs are found in plants.

In summary, the results of experiments presented above clearly identify AtCPT1 as an enzyme responsible for biosynthesis of long-chain dolichols in Arabidopsis. The mechanism of regulation of AtCPT1 activity in plant and yeast cells remains elusive and requires clarification. One might speculate that NgBR/NUS1-independent machinery might be needed to modulate the activity of AtCPT1 and affect the length of the synthesized polyprenol chain. This intriguing question will be a subject of future studies. 

## 3. Materials and Methods

### 3.1. Chemicals

All polyprenol and dolichol standards were from the Collection of Polyprenols of the Institute of Biochemistry and Biophysics (Warsaw, Poland). Components of growth media were purchased from BioShop (Burlington, ON, Canada). Chromatographic materials and HPLC solvents were from Merck (Darmstadt, Germany). All other chemicals were of practical grade (p.a.) quality and were purchased from Sigma-Aldrich Chemie GmbH (Poznan, Poland). 

### 3.2. Plant Materials and Growth Conditions

*Arabidopsis thaliana* ecotype Columbia 0 (Col-0) and At2g23410 insertion mutants *cpt1-1* (SALK_038151), *cpt1-2* (SALK_032276) and *cpt1-3* (SALK_100795) were obtained from the Nottingham Arabidopsis Stock Center (Loughborough, UK). 

Plants for genotyping were grown in potting soil in a greenhouse under a long-day (16 h light) photoperiod at 21/18 °C. Genotyping was performed with specific primers designed with the aid of T-primer design tool and LBb1.3 primer (Appendix A).

Plants for RNA isolation, RT-PCR and polyisoprenoid extraction were grown hydroponically in long day photoperiod (16 h day, 8 h night; 21 or 18 °C, respectively). The culture medium (Gibeaut’s solution) was prepared according to the literature [33].

Plants for root elongation measurements were sown on ½ Murashige and Skoog (MS) basal salt mixture medium (pH 5.7) supplemented with vitamin and solidified with 0.8% agar (*w*/*v*). Before sowing seeds were surface-sterilized in 20% (*w*/*v*) chlorine bleach and then washed with 70% (*w*/*v*) ethanol and sterile distilled water. After sowing seeds were allowed to imbibe in the dark at 4 °C for 2 days and were then grown on vertically oriented plates at 21/18 °C with a 16 h/8 h light/dark photoperiod.

*Nicotiana benthamiana* were cultivated in potting soil in a greenhouse under a long-day photoperiod. 

### 3.3. Yeast Materials

*Saccharomyces cerevisiae* strains used in this study:SS328 (wild type, MATα ade2-101 ura3-52his3Δ 200lys2-801) andYG932 (rer2Δ mutant, MATα rer2Δ::kanMX4 ade2-101 ura3-52his3Δ 200 lys-801) were kind gifts of Dr. C.J. Waechter (University of Kentucky College of Medicine, Lexington, KY, USA).YG938 (srt1Δ mutant, MATα Δ srt1::kanMX4 ade2-101 ura3-52 his3Δ200 lys2-801) was obtained from Dr. M. Aebi, (ETH, Zurich, Switzerland).

### 3.4. Preparation of AtCPT1 and LEW1 Expression Constructs

Total RNA was isolated from Arabidopsis Col-0 roots and purified using the RNeasy Plant Mini Kit (Qiagen, Hilden, Germany), transcribed to cDNA using SuperScript^®^ First-Strand Synthesis System (Invitrogen, Carlsbad, CA, USA), and AtCPT1 and LEW1 CDSs were amplified by PCR using specific primers (Appendix A). 

The purified PCR product of AtCPT1 was subcloned into the pENTR vector according to the manufacturer’s instructions (pENTR D-TOPO; Invitrogen). The AtCPT1 CDS was recombined from the pENTR vector into the destination vectors: yeast expression vector pYES-DEST52 or plant expression Gateway binary vectors [34] containing 35S promoters: ImpGWB402 (no tag) or ImpGWB451 (C-terminal G3GFP tag) using the Gateway LR Clonase enzyme mix (Invitrogen) according to the manufacturer’s protocol.

Coding regions of AtCPT1 and LEW1 were amplified by PCR using gene specific primers (Appendix A) and subcloned into pESC-URA yeast dual expression vector (Agilent, Santa Clara, CA, USA) according to the manufacturer’s protocol. The PCR product for AtCPT1 was digested with restriction enzymes *NotI* and *BcuI* and inserted into the corresponding site behind the Gal-10 promoter in the correct orientation. The PCR product for LEW1 was digested with *BamHI* and *SalI* and inserted into the corresponding site behind the Gal-1 promoter in the correct orientation.

### 3.5. Generation of Arabidopsis Plants Overexpressing AtCPT1

The S35::*AtCPT1* construct was introduced into *Agrobacterium tumefaciens* GV3101 strain and then used for transformation of *A. thaliana* (Col-0) wild type plants by the floral dip method [35]. F1 seeds were screened on ½ MS agar plates supplemented with 50 mg/L kanamycin (Sigma) and verified by PCR using specific primers for the S35 promoter and *AtCPT1*. F3 of two *CPT1-OE* lines (24 and 25) was used for the expression and lipid content analyses. 

### 3.6. Transient Expression of AtCPT1 in Nicotiana benthamiana

The S35::*AtCPT1*-GFP or S35::*AtCPT1* constructs, respectively, were introduced into *Agrobacterium tumefaciens* GV3101 strain. The Agrobacterium cultures were grown in liquid LB medium supplemented with 100 mg/L spectinomycin at 28 °C to an OD_600_ of 1.5, diluted to an OD_600_ of 0.4 in infiltration medium containing 10 mM MES (pH 5.6) 10 mM MgCl_2_ and 100 µM acetosyringone and infiltrated into the abaxial side of the *N. benthamiana* leaves. Infiltrated leaves (2 or 4 days post infiltration, dpi) were observed using a confocal microscope or harvested for polyisoprenoid extraction, respectively. 

### 3.7. Generation of Yeast Overexpressing AtCPT1

Preparation and transformation of competent *S. cerevisiae* cells was performed using *S. c*. EasyComp™ Transformation Kit (Invitrogen) according to manufacturer’s protocol. Transformant selection was performed on plates containing minimal medium without uracil (SC-U) with 2% glucose and G418 antibiotic for 3 days at 28 °C. To induce protein expression, the transformants were grown in SC-U medium with 2% galactose and G418. After 2–4 days of growth the yeast cells were collected for polyisoprenoid and CPY analyses. In vivo complementation assay of *rer2*Δ mutation with *AtCPT1* was performed as described previously [24].

### 3.8. Polyisoprenoid Extraction and Purification

Plant polyisoprenoid: 1.5 g of Arabidopsis or Nicotiana leaves were homogenized in a mixture of chloroform/methanol (C/M, *v*/*v*) supplemented with internal standard of Pren-14 (10 μg) for 48 h at room temperature. Extracted lipids were dissolved in 5 mL of a mixture containing toluene/7.5% KOH/95% ethanol (20/17/3, *v*/*v*/*v*) and hydrolyzed for 1 h at 95 °C, then extracted with hexane, purified on silica gel columns, dissolved in propan-2-ol and analyzed as described earlier [7].

Yeast polyisoprenoid: Yeast cells were harvested by centrifugation of 100 mL of yeast culture at 3300× *g* (Allegra; Beckman, Indianapolis, IN, USA) for 10 min at room temperature, washed with water and hydrolyzed for 1 h at 95 °C in 10 mL of solution containing 25% KOH and 65% ethanol supplemented with internal standard Pren-28 (10 μg). Subsequently, lipids were extracted with hexane, purified on silica gel columns, dissolved in propan-2-ol and analyzed as described earlier [36].

### 3.9. HPLC/UV Analysis of Polyisoprenoids

Lipids extracted from Arabidopsis roots were analyzed using 4.6 × 75 mm ZORBAX XDB-C18 (3.5 μm) reversed-phase column (Agilent, Santa Clara, CA, USA) and Waters Photodiode Array Detector (210 nm, Milford, MA, USA). 

Lipids extracted from yeast and Arabidopsis and tobacco leaves were analyzed using 4.6 × 50 mm Cortecs-C18 (2.7 μm) reversed-phase column (Waters) and UHPLC Waters Acquity Arc UV/Vis Detector (210 nm).

The external standards of a polyprenol mixture (Pren-9, 11–23, 25) and dolichol mixture (Dol-15 to Dol-24) were used to identify of chain length of polyisoprenoids. Pren-27 was used as internal standard for quantitative determination of polyisoprenoids. 

### 3.10. Confocal Microscopy

For subcellular localization analysis of AtCPT1 cd3-959 (ER m-Cherry), cd3-954 (ER CFP) and cd-967 (Golgi m-Cherry) vectors were used as organelle markers [37]. Confocal Images were taken under an Nikon C1 confocal system built on TE2000E with 408, 488 and 543 nm laser excitations for CFP (450/35 nm emission filter), GFP (515/30 nm emission filter), and m-Cherry (605/75 emission filter), respectively.

### 3.11. Statistical Analysis

Statistical analysis was performed using the data analysis tool Student’s *t* test in the Graph Pad Prism program (Graphpad software, version 5, San Diego, CA, USA). All experiments were repeated at least three times with similar trends.

### 3.12. Accession Numbers

Arabidopsis Genome Initiative locus identifiers for the genes mentioned in this article are as follows: *AtCPT1* (At2g23410), *AtCPT3* (At2g17570), *AtCPT6* (At5g58780), *AtCPT7* (At5g58770) and *AtCPT9* (At5g58784).

## Figures and Tables

**Figure 1 molecules-24-02789-f001:**
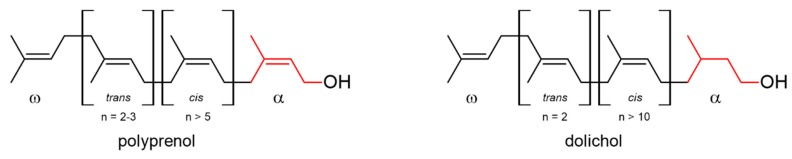
Structure of polyprenol and dolichol. α and ω represent terminal isoprene units. The internal isoprene residues in *trans* and *cis* configuration are indicated.

**Figure 2 molecules-24-02789-f002:**
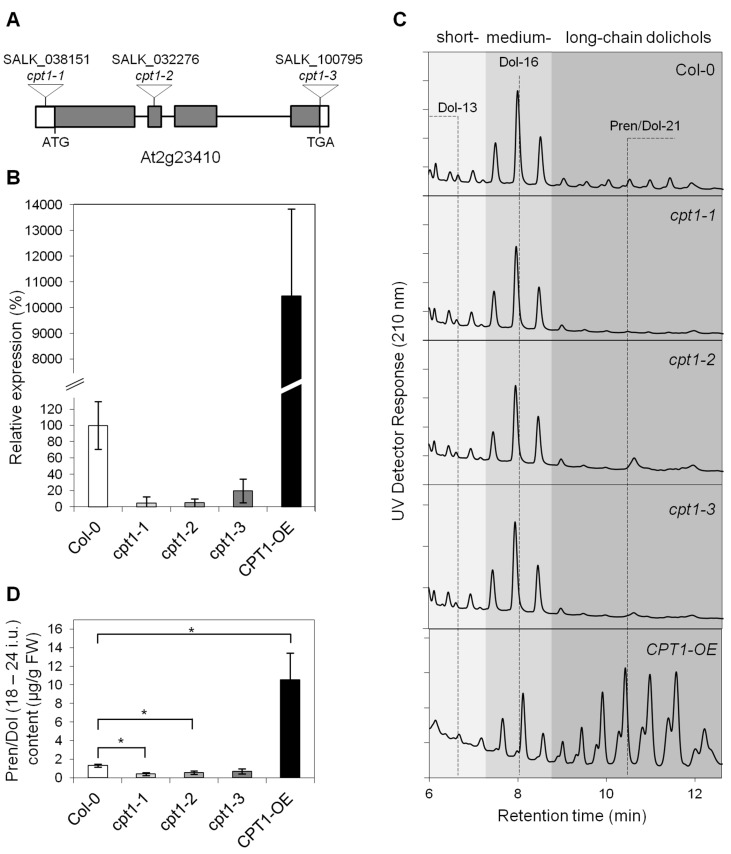
The effect of *AtCPT1* deficiency or overexpression on long-chain dolichols accumulation—in planta analysis. (**A**) *AtCPT1* gene structure. The start codon (ATG) and the stop codon (TGA) are indicated. Gray boxes indicate exons, and lines between boxes indicate introns. The T-DNA insertion sites in *cpt1-1, cpt1-2* and *cpt1-3* mutant lines are shown. (**B**) Relative expression of *AtCPT1* in Arabidopsis roots of the wild type (Col-0), three independent T-DNA insertion mutant lines (*cpt1-1*, *cpt1-2* and *cpt1-3*) and *AtCPT1* overexpressing line (*CPT1-OE*). AtCPT1 mRNA abundance was quantified by Real-Time PCR and is presented as a percentage of the wild type (100%). (**C**) Representative HPLC/UV chromatograms of total polyisoprenoids extracted from the roots of Arabidopsis wild typeT-DNA insertion mutants and CPT1-OE plants. The dominating homologues of polyprenols and dolichols accumulated in Arabidopsis roots, Dol-13, Dol-16 and Pren/Dol-21, respectively are indicated. (**D**) Quantification of the long-chain polyprenols and dolichols extracted from roots of studied plants. Data represents three independent experiments. Asterisk (*) indicates significant difference (0.01 < *P* < 0.05 in Student’s *t*-test) between WT and mutant plants.

**Figure 3 molecules-24-02789-f003:**
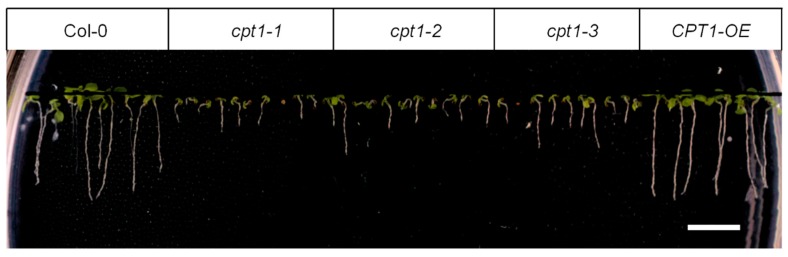
Growth phenotype of *AtCPT1* mutants. The wild type (Col-0), T-DNA insertion mutants (*cpt1-1*, *cpt1-2* and *cpt1-3*) and *AtCPT1* overexpressing (*CPT1-OE*) Arabidopsis seedlings were grown for seven days on vertically oriented agar plates. Scale bar = 1 cm. Representative data are shown.

**Figure 4 molecules-24-02789-f004:**
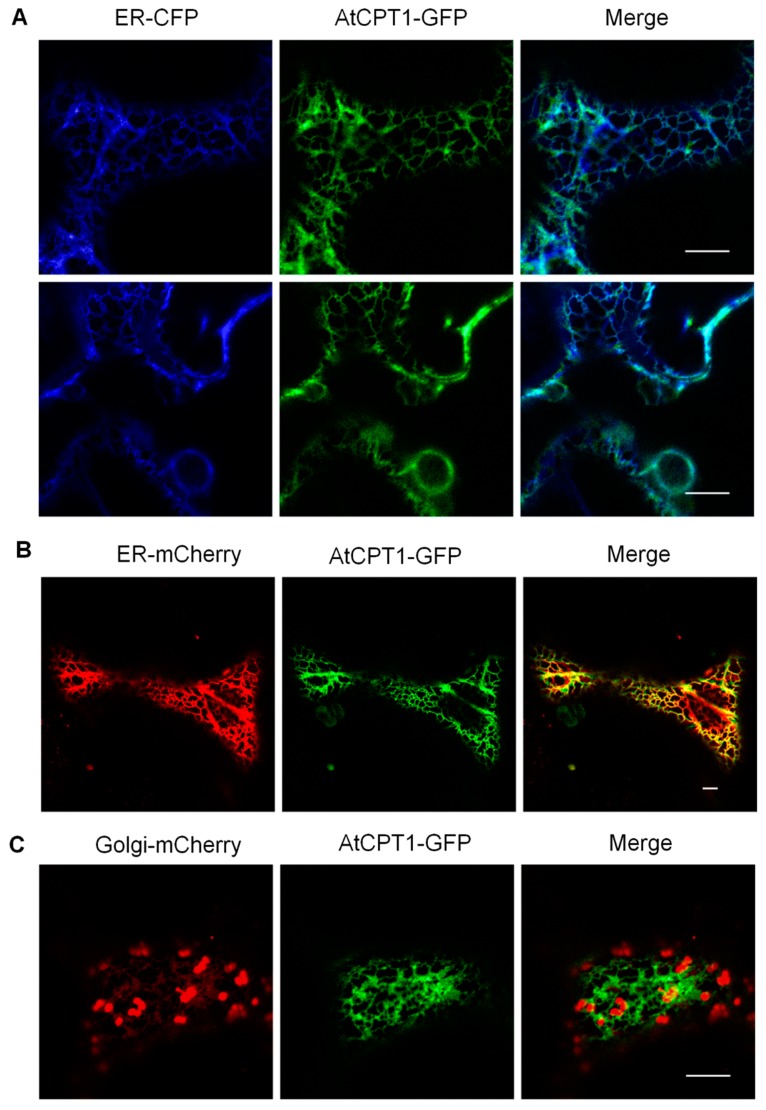
Subcellular localization of AtCPT1. Tobacco leaves transiently expressing *AtCPT1* containing a C-terminal GFP-tag fused at the C-terminus to GFP. AtCPT1-GFP signal is co-localized mostly with the markers of the endoplasmic reticulum (**A**,**B**) while no co-localization with Golgi marker (**C**) is observed. P_S35_::*AtCPT1*-GFP (green) and organelle markers ER-CFP (**A**), ER-mCherry (**B**) or Golgi-mCherry (**C**) were transiently expressed in *N. benthamiana* leaves by agroinfiltration. Scale bars = 10 µm. Representative pictures are shown.

**Figure 5 molecules-24-02789-f005:**
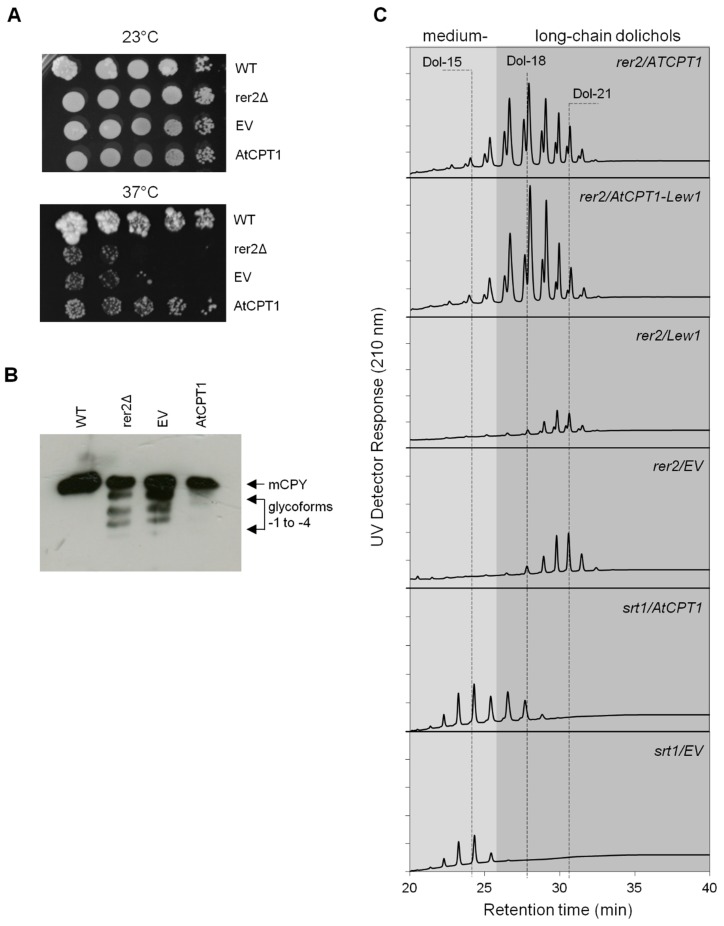
Analysis of enzymatic activity of AtCPT1 in the S. *cerevisiae* CPT mutants, *rer2*Δ or *srt1*Δ. (**A**) Expression of *AtCPT1* in *rer2*Δ mutant complements defects in growth. *rer2*Δ cells were transformed with the empty plasmid pYES-DEST52 and *AtCPT1* construct, respectively. Serially diluted yeast cultures were plated on solid YP-2% galactose medium and cultured for five days at 23 or 37 °C. (**B**) *AtCPT1* expression in *rer2*Δ yeast strain restores glycosylation status of carboxypeptidase Y (CPY). Protein extracts from yeast transformants were separated by SDS-PAGE and analyzed by Western blot with anti-CPY antibody. The positions of mature CPY (mCPY) and hypoglycosylated glycoforms lacking between one and four *N*-linked glycans (−1 to −4) are indicated. (**C**) Polyisoprenoid profiles of *rer2*Δ or *srt1*Δ yeast mutants transformed with empty plasmid, *AtCPT1, AtCPT1/Lew1* and *Lew1*, respectively. Presented are representative HPLC/UV chromatograms.

**Figure 6 molecules-24-02789-f006:**
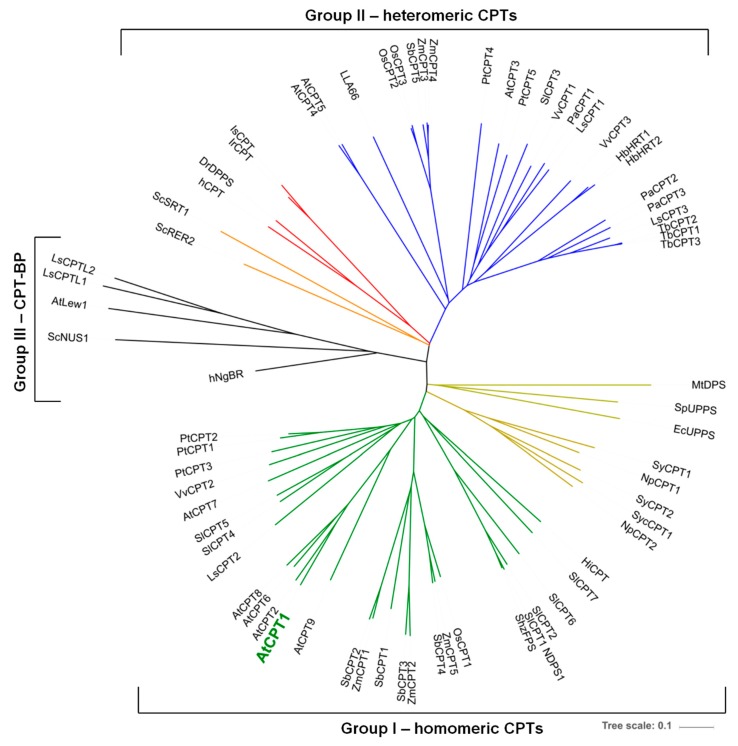
Phylogenetic analysis of CPTs. Neighbor-joining phylogenetic tree of CPTs and CPT related protein from plants (green or blue branches), animals (red), yeast (orange) and bacteria (yellow). The analyzed proteins sequences are divided into three groups: (I) the homomeric CPTs, (II) heteromeric CTPs that form a complex with a CPT-binding protein and (III) CPT-binding protein including the human NogoB receptor and its orthologs. Amino acid sequences were aligned using http://www.phylogeny.fr/index.cgi and the tree was constructed using https://itol.embl.de/. Species abbreviations: At, *Arabidopsis thaliana*; Dr, *Danio rerio*; Ec, *Escherichia coli*; Hb, *Hevea brasiliensis*; h, *Homo sapiens*; Hi, *Handroanthus impetiginosus*; Ir, *Ixodes ricinus*; Is, *Ixodes scapularis*; LL, *Lilium longiflorum*; Ls, *Lactuca sativa*; Mt, *Mycobacterium tuberculosis*; Np, *Nostoc punctiforme*; Os, *Oryza sativa*; Pa, *Parthenium argentatum*; Pt, *Populus trichocarpa*; Sb, *Sorghum bicolour*; Sc, *Saccharomyces cerevisiae*; Shz, *Solanum habrochaites*; Sl, *Solanum lycopersicum*; Sp, *Streptococcus pneumoniae*; Sy, Synechococcus; Syc, Synechocystis; Tb, *Taraxacum brevicorniculatum*; Vv, *Vitis vinifera*; Zm, *Zea mays*.

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
