# Peer review of "Long-Chain Polyisoprenoids Are Synthesized by AtCPT1 in *Arabidopsis thaliana"

_molecules, 2019, doi:10.3390/molecules24152789_

Round 1

Reviewer 1 Report

The authors identified a gene (AtCPT1) which was responsible for biosynthesis of long-chain dolichols in Arabidopsis roots. They investigated the function of AtCPT1 only through gene disruption and overexpression.

Major comments:

1, Since dolichols represent a large group of compounds, is it possible to identify the specific structure?

2, The authors showed the transcriptional levels of AtCPT1 in the gene disruption mutants and the AtCPT1 overexpressed strain. As we know, the transcriptional level could not be in accordance with the expression due to different translational efficiency. It is better to check the expression of AtCPT1 by Western blot analysis.

3, In Fig.1, many peaks appeared when AtCPT1 was overexpressed in Arabidopsis. Please identified them besides three peaks labeled. At least, the author should give the information or analysis.

4, It is interesting that the cpt1 mutants exhibited growth defects, could the authors give more analysis?

Author Response

We would like to thank all Reviewers for their valuable comments which stimulated our effort to improve the manuscript. Suggestions of the Reviewers have been carefully considered and implemented in the revised manuscript. The text has been partially re-written. New Figure 1 (Structures of polyprenol and dolichol) has been added. We have carefully checked the English writing and corrected the mistakes. Below please find a point-to-point list of answers to the issues raised in the Reviewers’ opinions.

Reviewer 1

The authors identified a gene (AtCPT1) which was responsible for biosynthesis of long-chain dolichols in Arabidopsis roots. They investigated the function of AtCPT1 only through gene disruption and overexpression.

Major comments:

1, Since dolichols represent a large group of compounds, is it possible to identify the specific structure?

Answer: Thank you for this comment. In this paper we are focusing on the functional analysis of the particular cis-prenyltransferase. The identification of specific structure of dominating dolichol produced by AtCPT1 is possible using mass spectrometry and NMR methods. MS methods has been used here and it confirmed Dol structure. NMR method requires a large amount (1 mg) of highly purified product. Moreower separation of dolichol from the accompanying polyprenol seems not feasible.

2, The authors showed the transcriptional levels of AtCPT1 in the gene disruption mutants and the AtCPT1 overexpressed strain. As we know, the transcriptional level could not be in accordance with the expression due to different translational efficiency. It is better to check the expression of AtCPT1 by Western blot analysis.

Answer: We agree that the Western blot analysis would be a good additional control that the AtCPT1 level is reduced in the cell. However, we have a good correlation of reduced transcript level and reduced content of the lipid products. Furthermore, we do not have an antibody directed specifically against AtCPT1. In the past, we have generated (Agrisera, Sweden) several antibodies directed against the Arabidopsis CPTs, but only one of them, against AtCPT6 (Surmacz et. al, BBA, 2014), specifically recognized target protein, the remaining antibodies showed cross reactions with the other Arabidopsis CPTs. 

3, In Fig.1, many peaks appeared when AtCPT1 was overexpressed in Arabidopsis. Please identified them besides three peaks labeled. At least, the author should give the information or analysis.

Answer: Upon overexpression of AtCPT1 in Arabidopsis we observed larger amounts of polyprenols that accompanied corresponding dolichols. Probably these increased amout of polyprenols is not efficiently reduced to dolichols by polyprenol reductase.

We changed a caption of this Figure in the revised manuscript. The sentence “The dominating homologues of polyprenols and dolichols accumulated in Arabidopsis roots, Dol-13, Dol-16 and Pren/Dol-21, respectively are indicated”

is changed to

“ The short- (Dol-12 to -14, dominating Dol-13), medium- (Dol-15 to -17, dominating Dol-16) and long-chain dolichols (Dol-19 to -30, dominating Dol-21) accumulated in Arabidopsis roots are indicated. Dolichols are accompanied by various amounts of corresponding polyprenols, especially in CPT1-OE plants.” The comment that polyprenol products of AtCPT1 are not efficiently reduced to dolichols was already mentioned in the original text.

4, It is interesting that the cpt1 mutants exhibited growth defects, could the authors give more analysis?

Answer: Explanation of the growth defects of cpt1 mutants requires further experiments and observations (e.g. microscopic observations, aplication of various growth conditions) and will be addressed in future work.

Reviewer 2 Report

Liliana Surmacz and colleagues presented the results of research showing that CPT1 is responsible for the synthesis of long-chain polyisoprenoids in A. thaliana.

The reviewed work is unique in that the researchers describe the location and function of AtCPT1, which has not been previously described in literature.

I have a few minor suggestions:

Page 2, line 73: correct name of the plant is „Lilium longiflorum”

Page 10, line 260: correct name of the bacteria is „Streptococcus pneumoniae”

Page 11, lines 291-292: I am not sure if in the sentence: „Nicotiana benthamiana were cultivated in potting soil in a greenhouse under a long-day photoperiod” is missing only a dot at the end of the sentence or there is a missing part of the text.

Introduction of the manuscript could be enriched by the structural formulas of dolichols and polyprenols.

Page 12, line 344: „Yeast cells (100ml) were harvested by centrifugation” - Should the sentence be: "Yeast cells were harvested by centrifugation of 100 ml of yeast culture at..."? It seems to me that 100 ml of the yeast cells were not used to extraction of polyisoprenoids.

Materials and Methods section lacks a description of the statistical methods used. Student's t-test is mentioned in Figure 1 caption.

In my opinion figure captions are too long. According to Instructions for Authors (https://www.mdpi.com/journal/molecules/instructions) all Figures, Schemes and Tables should have a short explanatory title and caption. Some sentence should be missed or should be transferred to Discussion or Materials and Methods sections, i.e.: Page 9, lines 210-211: "Note the accumulation of dolichol family composed of 15 to 20 211 isoprene units with dominating Dol-18 in rer2Δ and Dol-17 in srt1Δ cells, respectively."

Apart from the minor remarks mentioned above, the manuscript is interesting, introduction provide sufficient background, and results and discussion are decribed well.

Author Response

We would like to thank all Reviewers for their valuable comments which stimulated our effort to improve the manuscript. Suggestions of the Reviewers have been carefully considered and implemented in the revised manuscript. The text has been partially re-written. New Figure 1 (Structures of polyprenol and dolichol) has been added. We have carefully checked the English writing and corrected the mistakes. Below please find a point-to-point list of answers to the issues raised in the Reviewers’ opinions.

Reviewer 2

Liliana Surmacz and colleagues presented the results of research showing that CPT1 is responsible for the synthesis of long-chain polyisoprenoids in A. thaliana.

The reviewed work is unique in that the researchers describe the location and function of AtCPT1, which has not been previously described in literature.

 I have a few minor suggestions:

Page 2, line 73: correct name of the plant is „Lilium longiflorum”

Answer: The correct name of  this plant is placed in the revised text.

Page 10, line 260: correct name of the bacteria is „Streptococcus pneumoniae”

Answer: The correct name of  this bacteria is placed in the revised text.

Page 11, lines 291-292: I am not sure if in the sentence: „Nicotiana benthamiana plants were cultivated in potting soil in a greenhouse under a long-day photoperiod” is missing only a dot at the end of the sentence or there is a missing part of the text.

Answer: A dot was missing, now it is added to the revised text.

Introduction of the manuscript could be enriched by the structural formulas of dolichols and polyprenols.

Answer: The figure showing structure of dolichol and polyprenol is added as Figure 1 to Introduction section in the revised manuscript.

Figure 1. Structure of polyprenol and dolichol. The isopren residues in trans and cis configuration are indicated. α and ω represent terminal isopren units.

Page 12, line 344: „Yeast cells (100ml) were harvested by centrifugation” - Should the sentence be: "Yeast cells were harvested by centrifugation of 100 ml of yeast culture at..."? It seems to me that 100 ml of the yeast cells were not used to extraction of polyisoprenoids.

Answer: This sentence is changed to “Yeast cells were harvested by centrifugation of 100 ml of yeast culture at….” in the revised text as suggested by the Reviever.

Materials and Methods section lacks a description of the statistical methods used. Student's t-test is mentioned in Figure 1 caption.

Answer: The paragraph “Statistical analysis” is added to Materials and Methods section in the revised manuscript.

4.1 Statistical analysis

Statistical analysis was performed using the data analysis tool - Student’s t-test in the Graph Pad Prism program (Graphpad software). All experiments were repeated at least three times.

In my opinion figure captions are too long. According to Instructions for Authors (https://www.mdpi.com/journal/molecules/instructions) all Figures, Schemes and Tables should have a short explanatory title and caption. Some sentence should be missed or should be transferred to Discussion or Materials and Methods sections, i.e.: Page 9, lines 210-211: "Note the accumulation of dolichol family composed of 15 to 20 211 isoprene units with dominating Dol-18 in rer2Δ and Dol-17 in srt1Δ cells, respectively."

Answer: The figure captions are changed in the revised manuscript.

Figure 2. Phenotypes of the wild type (Col-0), T-DNA insertion mutants (cpt1-1cpt1-2 and cpt1-3) and AtCPT1overexpressing (CPT1-OE) Arabidopsis seedlings at 7 day after germination. Plants were grown on vertically oriented agar plates. cpt1 mutants show reduced growth and root length in comparison to Col-0 and CPT1-OE plants. Scale bar = 1 cm. Representative data are shown.

is changed to

Figure 2. Growth phenotype of AtCPT1 mutants. The wild type (Col-0), T-DNA insertion mutants (cpt1-1cpt1-2and cpt1-3) and AtCPT1overexpressing (CPT1-OE) Arabidopsis seedlings were grown for 7 days on vertically oriented agar plates. Scale bar = 1 cm. Representative data are shown.

Figure 3. The endoplasmic reticulum (ER) localization of the protein encoded by AtCPT1 containing an C-terminal GFP-tag in tobacco leaves visualized by confocal immunofluorescence. AtCPT1-GFP signal is co-localized mostly with the markers of the endoplasmic reticulum (A, B) while no co-localization with Golgi marker (C) is observed. PS35::AtCPT1‐GFP (green) and organelle markers ER-CFP (A), ER-mCherry (B) or Golgi-mCherry (C) were transiently expressed in N. benthamiana leaves by agroinfiltration. Scale bars = 10 µm. Representative pictures are shown.

is changed to

Figure 3. Subcellular localization of AtCPT1.Tobacco leaves transiently expressing AtCPT1 containing an C-terminal GFP-tag fused at the C-terminus to GFP. AtCPT1-GFP signal is co-localized mostly with the markers of the endoplasmic reticulum (A, B) while no co-localization with Golgi marker (C) is observed. PS35::AtCPT1‐GFP (green) and organelle markers ER-CFP (A), ER-mCherry (B) or Golgi-mCherry (C) were transiently expressed in N. benthamiana leaves by agroinfiltration. Scale bars = 10 µm. Representative pictures are shown.

 Answer: The sentence “Note the accumulation of dolichol family composed of 15 to 20 isoprene units with dominating Dol-18 in rer2Δ and Dol-17 in srt1Δ cells, respectively.” is removed from the revised manuscript.

Apart from the minor remarks mentioned above, the manuscript is interesting, introduction provide sufficient background, and results and discussion are decribed well.

Reviewer 3 Report

This is a solid and interesting study. I believe that it will be of sufficient road and specialist interest to merit publication. There area few minor typos, e.g. stabile for stable but generally well written and produced.

Author Response

Thank you very much for your opinion.